# Molecular Events of Rice AP2/ERF Transcription Factors

**DOI:** 10.3390/ijms231912013

**Published:** 2022-10-10

**Authors:** Wei Xie, Chaoqing Ding, Haitao Hu, Guojun Dong, Guangheng Zhang, Qian Qian, Deyong Ren

**Affiliations:** State Key Lab of Rice Biology, China National Rice Research Institute, Hangzhou 310006, China

**Keywords:** rice, AP2/ERF, classification and feature, gene function, regulatory mechanism, breeding potential

## Abstract

APETALA2/ethylene response factor (AP2/ERF) is widely found in the plant kingdom and plays crucial roles in transcriptional regulation and defense response of plant growth and development. Based on the research progress related to AP2/ERF genes, this paper focuses on the classification and structural features of AP2/ERF transcription factors, reviews the roles of rice AP2/ERF genes in the regulation of growth, development and stress responses, and discusses rice breeding potential and challenges. Taken together; studies of rice AP2/ERF genes may help to elucidate and enrich the multiple molecular mechanisms of how AP2/ERF genes regulate spikelet determinacy and floral organ development, flowering time, grain size and quality, embryogenesis, root development, hormone balance, nutrient use efficiency, and biotic and abiotic response processes. This will contribute to breeding excellent rice varieties with high yield and high resistance in a green, organic manner.

## 1. Introduction

Transcription factors (TFs) are a group of protein molecules that can specifically bind to cis-acting elements in the promoter region of eukaryotic genes to ensure that target genes are expressed at a specific intensity in a specific time and space, thus playing important roles in plant growth, development and environmental responses. In plants, about 60 kinds of transcription factors have been identified, such as AP2/ERF, bZIP, C2H2, MYB, MADS, NAC, and WRKY [1]. TFs and various DNA binding domains control various regulation processes in higher plants. AP2/ERF is one of the evolutionary oldest and largest transcription factor families in plants. It was first isolated from *Arabidopsis thaliana* by Jofuku in 1994 and is widely involved in plant growth and development processes, including plant height, root development, flower development, seed germination, organ size, resistance to biotic and abiotic stress, and plant hormone signal transduction pathways [1,2]. As an important food crop, rice (*Oryza sativa*) encounters pests, diseases, and abiotic stresses such as extreme temperatures, drought and flooding, during the life cycle, which greatly affects the rice production. Studies have shown that rice AP2/ERF transcription factors play crucial roles in growth and development, and also play important roles in improving the resistance of rice to various stresses, pests and diseases.

## 2. Classification, Features and Binding Elements

In the plant kingdom, the AP2/ERF family is a large family of transcription factors with conserved AP2/ERF domains composed of about 60–70 amino acid residues [3,4,5,6]. According to the number of AP2/ERF domains and the different functions of genes, the superfamily can be divided into four families: AP2, ERF, RAV, and Soloist [7,8,9,10]. The AP2 family has two duplicated AP2/ERF domains, which can be further subdivided into AP2 and ANT subfamilies [11]. ERF family proteins have only one AP2/ERF domain, which can be subdivided into two subfamilies, ERF and CBF/DREB, according to the sequence of the DNA binding domain. The proteins encoded by ERF subfamily genes bind to AGCCGCC core motifs, while CBF/DREB subfamily members recognize the cis-acting elements A/GCCGAC, named the C-repeat [12,13,14]. ERF and DREB proteins are associated with abiotic stress in plants and are affected by drought, low temperature and high salt. In addition to the AP2/ERF domain, RAV family proteins possess a VP1/ABI3 binding domain. These proteins are negative regulators of growth and development in arabidopsis and play key regulatory roles in plant defense pathways. Phylogenetic analysis of AP2/ERF TFs in rice showed that alanine 14 (Ala-14) and aspartic acid 19 (Asp-19) of ERF subfamily proteins are conserved among the members containing a single AP2/ERF domain, while CBF/DREB subfamily proteins contained valine (Val-14) at the 14th and glutamine (Glu-19) at the 19th positions [9,15]. Compared with the conserved Arg-6 of a single domain protein, Thr-6 was more conserved among the proteins containing two AP2/ERF domains [9,15]. It was also found that there is histidine in each AP2 domain of the AP2 subfamily proteins with two AP2 domains, but not in the other AP2/ERF proteins with one AP2 domain [9,15]. There are 165 AP2/ERF TFs in rice, including 24 AP2 subfamily members, 136 ERF (ERF/DREB) subfamily members, five RAV subfamily members, and zero Soloist subfamily members [1,8] (Figure 1, Table 1).

## 3. Molecular Roles of AP2/ERF TFs in Rice

Numerous reports have documented that rice AP2/ERF TFs are important regulators involved in plant growth and hormonal regulation, including spikelet development, grain characteristics, root development and formation, drought tolerance, salinity tolerance, temperature tolerance, and nutrient use [1,16,17]. Here, we review the functions and the regulatory networks of AP2/ERF TFs and discuss potential applications in rice breeding.

### 3.1. Spikelet Determinacy and Organ Development

The spikelet, as a grass-specific basal inflorescence unit, can be divided into two types: determinate spikelet (DS) and indeterminate spikelet. In rice with DS, the spikelet meristem produces two lateral floral meristems and one terminal floral meristem (TFM). Two LFMs form one pair of sterile lemmas, and the TFM develops one terminal floret, which generates one seed [18,19]. 

So far, few genes have been reported to maintain spikelet meristem fate, but AP2/ERF genes have been found to be involved in this process [17,18]. AP2/ERF family genes play an important role in the regulation of spikelet meristem determinacy and organ identity (Figure 2) [18,19]. *FRIZZY PANICLE (FZP)* delays the transition from spikelet meristem to flower meristem, determines the identity of sterile lemma, and is required for the formation of axillary meristem in the spikelet meristem and floral meristem [19,20]. Loss-of-function of *FZP* results in extra rudimentary glumes and the absence of sterile lemmas and florets on the rachilla position [19]. In weak allelic mutants *fzp-12* and *fzp-13*, the sterile lemmas are converted to rudimentary glumes, while the sterile lemmas are converted to lemmas (degraded florets) in *FZP* over-expression plants [19]. *SUPERNUMERARY BRACT (SNB)* is necessary for the correct transition from spikelet meristem to floral meristem and the correct formation of floral organ pattern [21]. In the *snb* mutant, the rudimentary glume-like and lemma/palea-like organs are generated, and the sterile lemmas fail to develop [21]. Meantime, ectopic lodicules are elongated and occasionally form on the palea side, the stamens were reduced, and additional pistils were formed [21]. In a few cases, extra florets are produced in the axils of sterile lemma [21]. In the *Oryza sativa indeterminate spikelet (osids1)* mutant, sterile lemmas are replaced by rudimentary glumes [21]. The *osids1**/snb* mutant develops more rudimentary glumes, lemmas, and paleae than the single mutant. Similarly, the *osids1**/snb* mutant shows abnormal floral organs, including lemma/palea-like lodicules, fewer stamens, lodicule-stamen chimera and stamen-pistil chimera organs [22]. Together, *SNB* and *OsIDS1* co-regulate spikelet development and organ identity and play a redundant role in the spikelet meristem transition and determinacy. Moreover, *SNB* and *OsIDS1* cooperatively participate in the determination of floral meristem fate in a dose-dependent manner. *MULTI-FLORET SPIKELET 1 (MFS1)* determines the spikelet meristem determinacy and floral organ identity, and positively regulates the expressions of spikelet development-related gene *G1* and *IDS*-like genes *SNB* and *OsIDS1* [18]. In the *mfs1* mutant, the transition from spikelet meristem to floral meristem is delayed, resulting in extra lemmas and paleae (degraded florets), and degraded sterile lemmas and paleae [18]. These results suggest that AP2/ERF genes inhibit spikelet indeterminacy, and determine organ fate, and it is possible to breed rice cultivars with multi-floret spikelets by altering meristem determinacy and/or inducing lateral florets, thereby increasing the grain number per panicle (Figure 2, Table 2).

### 3.2. Flowering Regulation

Flowering time is a key agronomic trait affecting rice yield and quality. *OsIDS1* and *SNB* are involved in the regulation of flowering in rice (Figure 2, Table 3). Over-expression of *OsIDS1* or *SNB* delays rice flowering. In over-expression plants, two florigen genes *Heading date-3a (Hd3a)* and *Rice Flowering Lous T1 (RFT1)*, and their direct upstream regulator *Early heading date 1 (Ehd1)*, were inhibited [22]. MiR172d promoted rice flowering by reducing the expression of *SNB* and *OsIDS1*. The expression levels of *SNB* and *OsIDS1* were positively regulated by *CONSTANS-LIKE 4 (OsCOL4)* and *Heading date 1 (Hd1)*, and they inhibited floral transition by repressing *Ehd1* expression [23]. This showed that the AP2 subfamily *IDS*-like genes *OsIDS1* and *SNB* delay rice flowering (Figure 2), and suggest creation of rice new germplasms with early flowering and early maturing can be achieved by editing *IDS*-like genes.

### 3.3. Role in Grain Size, Quality and Shattering

Grain properties such as size, quality and shattering determine final edible production and the degree of mechanized harvesting. *SNB* influences grain size and weight by regulating cell division and elongation and participating in both brassinosteroid signaling and auxin signaling pathways [24]. CRISPR/Cas9 knockdown of *SNB* plants showed an increase in grain length, width and 1000-grain weight, while *SNB* over-expression decreased grain length, width and 1000-grain weight. *SNB* is also involved in regulating grain shattering. The allele *SUPPRESSION OF SHATTERING 1 (SSH1*) of *SNB* affects the deposition of lignin and the development of abscission zone (AZ) by positively regulating the expression of *SEED SHATTERING 1 (qSH1)* and *SEED SHATTERING 5 (SH5)* genes and regulates rice grain shattering. The *ssh1* gene has the genetic effect of increasing grain length and grain weight, and the introduction of *ssh1* into the excellent *indica* rice variety 93–11 further increased grain length and weight, indicating that the *ssh1* gene has the potential to improve rice yield [25]. *SEED SHATTERING ABORTION 1 (SHAT1)*, a key gene for rice grain shattering, is continuously expressed in AZ at early spikelet development stages by participating in AZ development. *SEED SHATTERING 4 (SH4)* and *qSH1* are also necessary to reduce grain shattering during rice domestication. Genetic analysis showed that *SHAT1* expression in AZ was positively regulated by *SH4*. *qSH1* acts downstream of *SHAT1* and *SH4*, and promotes AZ differentiation by maintaining the expression of *SHAT1* and *SH4* in AZ, thereby controlling grain shattering [26]. As a transcriptional repressor, OsERF115 interacts with ETHYLENE-INSENSITIVE 3-LIKE 1 (OsEIL1) to regulate grain size [27]. Haplotype analysis showed that the SNP variation of the *OsERF115* promoter that can be bound by ETHYLENE INSENSITIVE 3 (EIN3), was significantly correlated with *OsERF115* expression level and grain weight, suggesting that natural variation of the *OsERF115* promoter contribute to grain size diversity. Over-expression of *OsERF115* promotes longitudinal cell elongation and lateral cell division of the hull, enhances grain filling, and increases grain length, width, thickness, and weight, whereas knockdown of *OsERF115* does the opposite, indicating that *OsERF115* positively regulates grain size and weight [27]. *FZP* was also reported to positively regulate grain size. The strong allele mutant of *fzp* could not form complete spikelets or floral organs, while the weak *fzp-12* mutant produced smaller grains [19]. Another allelic mutant *small grain and dense panicle 7 (sgdp7)* has an inserted 18-bp repeat in the upstream promoter of *FZP* gene, which inhibits the expression of *FZP*, prolongs spikelet branching, increases spikelet number per panicle, and decreases grain weight [28]. *SALT-RESPONSIVE ERF 1 (SERF1)* negatively regulates grain size and filling by affecting *RICE PROLAMIN-BOX BINDING FACTOR (RPBF)* expression. Loss of *SERF1* function increases the expression of *RPBF*, leading to an increase in grain size and starch content, while *SERF1* over-expression decreases the expression of *RPBF*, leading to a decrease in grain size [29]. 

Starch is the main component of endosperm, and its content and structure directly affect grain quality. *RICE STARCH REGULATOR 1* (*RSR1*) encodes a AP2/ERF transcription factor, the loss function of *rsr1* increases amylose content and changes amylopectin structure [30]. The changed starch grains have reduced gelatinization temperature. Moreover, the *rsr1* mutant grains become larger and yield increases. In *RSR1* over-expression plants, the structure of amylopectin and starch gelatinization characteristics both change in the opposite trend, indicating that *RSR1* regulates starch synthesis in rice grains [31]. 

Taken together, AP2/ERF genes play key roles in determining grain size, quality and shattering, thereby influencing grain yield and quality (Table 4, Figure 2). Exploring AP2/ERF genes and excellent alleles will further reveal genetic mechanism and creating a series of new germplasms through gene pyramiding and transgenic methods will provide a blueprint for breeding new rice varieties with appropriate grain size and good quality, thereby achieving the goal of high yield and quality.

### 3.4. Role in Embryogenesis

Apomixis includes parthenogenesis, apogamety and apospory. In apomictic plants, seed formation does not undergo exchange during meiosis, and the offspring retain the mother genotype, which plays an important role in heterosis fixation. The *BABY BOOM* (*BBM*) genes of the AP2/ERF family are involved in the initiation of rice embryogenesis (Figure 3, Table 5). Ectopic expression of *BBM1* in fertilized eggs can induce the production of somatic embryos, and ectopic expression in egg cells can induce parthenogenesis [31]. In the transgenic plant BBM1-ee with specific expression of *BBM1* in egg cells, the embryo structure could be observed, but the endosperm was lacking, and the seeds were aborted, suggesting that *BBM1* could still induce embryo formation without spermatogenesis. Synthetic apomictic (S-APO) plants were further generated by using a mitosis instead of a meiosis technique (MiMe, knockdown of *OSD1*, *PAIR1* and *REC8*) in BBM1-ee transgenic plants. It was found that S-APO plants could successfully induce parthenogenesis through seeds [31]. The *bbm1/bbm2/bbm3* mutant had no obvious vegetative growth and flower defects and could eventually form normal seeds. These results indicate that *BBM2* and *BBM3* have redundant functions compared to *BBM1*. These data show that AP2/ERF genes can induce parthenogenesis, making it possible to propagate rice seeds asexually (Figure 3) [31].

### 3.5. Root Initiation and Formation

Plant roots mainly absorb water and nutrients, metabolize, are involved in nitrogen fixation, and reproduce. The growth has a complex regulatory system [32]. AP2/ERF family genes, such as *Crown rootless 5 (Crl5)*, *OsERF48* and *ERF3,* are involved in root formation and development. *Crl5* encoded as a nitrpoERF transcription factor promotes crown root development, and its expression is induced by auxin, thereby positively regulating the expression of *RESPONSE REGULATOR 1 (OsRR1)* that inhibits the cytokinin signal [33]. The *crl5* mutant produces a small number of crown roots, and the initiation of crown root primordia is destroyed. *Crl5* over-expression plants do not induce ectopic roots, but their callus forms adventitious roots. The promoter of *Crl5* has an auxin responsive element, and driving the over-expression of *OsRR1* by the promoter restores the defect of adventitious roots formation in the *crl5* mutant. *CROWN ROOTLESS 1 (CRL1)* is the essential factor in regulating the crown root. The *crl1/crl5* mutant shows additive effects, indicating that *Crl1* and *Crl5* regulate the occurrence of adventitious roots through two different genetic pathways. The *radicleless 1 (ral1)* mutant cannot produce seed roots but can form crown roots. The *crl1/ral1* mutant has an additive phenotype, indicating that the occurrence of seed roots and crown roots is regulated by different genetic mechanisms in rice [34]. In the process of root development, *ETHYLENE-RESPONSE AP2/ERF FACTOR (OsERF48)/DROUGHT RESPONSIVE AP2/EREBP 1 (OsDRAP1)* enhance root growth by regulating the expression of the calmodulin gene *CALMODULIN 16*
*(OsCML16)*. *OsERF48* over-expression plants have longer initial roots, more lateral roots and increased dry weight of roots, and the lateral roots of RNAi plants are reduced [35]. *OsERF3*/*OsAP37* regulates crown roots development and the expression of auxin and cytokinin response genes [36]. ERF3 directly binds to the promoter of the cytokinin response gene *RR2*, thereby positively regulating the expression of *RR2* and controlling the initiation of crown roots. In contrast, ERF3 interacts with the WUSCHEL-related homeobox 11 (WOX11) to inhibit the expression of *RR2*, thereby promoting crown root elongation [36]. The number of crown roots is decreased, and the primary roots become shorter in *ERF3* knockout plants, while the opposite is true for over-expression lines [36]. These AP2/ERF genes ae involved in root development and formation by complex regulatory pathways (Figure 4, Table 6). 

### 3.6. AP2/ERF Regulatory Roles Mediated by Hormones

Plant hormones, including auxin, gibberellin, cytokinin, abscisic acid, ethylene and brassinosteroids, play important roles in regulating cell growth and differentiation, organogenesis and abscission, flowering, senescence and maturation. 

Ethylene is an important phytohormone for plant growth, development and stress tolerance, and its biosynthesis is regulated by the AP2/ERF transcription factor. The drought-responsive ERF gene, *OsDERF1,* negatively regulates ethylene synthesis by activating the transcription of *OsERF3* and *OsAP2-39*, and plays a negative role in drought stress [37]. *SNORKEL1* and *SNORKEL2* are strongly induced by ethylene and derived internode elongation through the GA pathway [38]. Under deep water conditions, ethylene is accumulated in both deep rice and non-deep rice; ethylene accumulated in deep rice can induce the expression of *SNORKEL1* and *SNORKEL2* and promote internode elongation. However, non-deep rice cannot elongate internodes due to the deletion of these two genes. *OsERF2*/*WR4* is a downstream component of the ethylene signaling pathway and affects the expression of auxin and cytokinin signaling pathway-related genes, ABA accumulation and response, and plays a negative regulatory role in rice root growth [39]. Ethylene and gibberellin synergistically regulate internode development, which determines plant architecture. The ERF transcription factor *OsEATB*, cloned from indica 93–11, mediates crosstalk between ethylene and gibberellin to repress internode elongation [40]. *OsEATB* expression is negatively regulated by ethylene, ABA and abiotic stresses. Over-expression of *OsEATB* decreases the endogenous GA level, and *OsEATB* limits the ethylene-induced GA response by down-regulating the expression of GA biosynthesis gene *ent-kaurene synthase A* during internode elongation [40]. 

GA is the main phytohormone for regulating plant height and its biosynthesis can be regulated by the AP2/ERF family gene. *REDUCED PLANT HEIGHT1 (OsRPH1)* negatively regulates plant height, internode length, and leaf sheath length by controlling the expression of GA-related genes. Exogenous GA3 treatment restores the defect of plant growth in over-expression lines [41]. *OsAP2-39* controls ABA/GA balance, which regulates plant growth and seed production [42]. Over-expression of *OsAP2-39* causes a decrease of plant height, tillering number, leaf number and grain number per panicle, and delays heading. Exogenous GA can restore the phenotype of seedling height, germination rate and other defects at heading stage in over-expression lines. Exogenous ABA can aggravate the defects of delayed germination and growth in the over-expression lines. A recent study showed that a AP2 transcription factor, *SMALL ORGAN SIZE1 (SMOS1)*/*NITROGEN-MEDIATED TILLER GROWTH RESPONSE 5 (NGR5)* was a key element of gibberellin signaling pathway and interacted with gibberellin receptor GA-INSENSTIVE DWARF1 (GID1) protein. NGR5 can also interact with the protein complex of Polycomb Repressive Complex 2 (PRC2) to regulate the expression of target genes by mediating the methylation level of histone H3K27me3. Gibberellin can reduce epigenetic modifications by promoting the degradation of the NGR5 protein, thereby enhancing the transcriptional activity of target genes to achieve gibberellin inhibition of rice growth and development [43]. RLA1/SMOS1 interacts with and is phosphorylated by GSK2, an importance regulating factor in BR signaling pathway, thereby reducing its stability. RLA1/SMOS1 is also involved in the regulation of *OsBZR1*, a positive regulator of BR signaling pathway [44]. SMOS1 interacts with a member of the GRAS family gene *SMOS2*/*DWARF AND LOW-TILLERING (DLT)*, to form a key complex in auxin-brassinosteroids signaling crosstalk, and synergistically enhances transcriptional trans-activation activity [45].

AP2/ERF transcription factor HAIRY LEAF 6 (*HL6*) regulates auxin biosynthesis. *HL6* controls the elongation of epidermis hair, and its regulatory role in the elongation of epidermis hair is mainly dependent on the function of OsWOX3B [46]. The protein complex of HL6 and OsWOX3B enhances the binding ability of HL6 to the auxin-related gene *OsYUCCA5*. The surfaces of leaves, leaf sheaths and hulls of *HL6* over-expression plants ae covered with trichomes/hairs, and the content of indole-3-acetic acid (IAA) in leaves is increased, whereas, the number of long hairs is decreased in the *hl6* mutant. In addition, *HL6* regulates hair elongation in a dose-dependent manner, and a higher *HL6* expression leads to longer hairs. These studies show that AP2/ERF genes induce hormone responses by activating target genes or regulating various growth processes of rice as response factors (Figure 4, Table 7). 

### 3.7. Regulation of Nutrient Use Efficiency

If rice is short of nitrogen, it grows slowly, with short plants and yellow leaves, while excessive nitrogen fertilizer induces more leaves, easy lodging, decreased yield and quality, and serious environmental pollution. Improving the efficiency of nitrogen fertilizer utilization, reducing the input of chemical fertilizer and environmental pollution while ensuring the continuous increase of grain yield is a new breeding strategy. As AP2/ERF transcription factors, *OsDREB1C* and *NGR5* increased rice nitrogen use efficiency. *OsDREB1C* can simultaneously improve the photosynthetic efficiency and nitrogen use efficiency of rice, leading to early heading, high yield and early maturity, thereby increasing rice yield [47]. Over-expression of *OsDREB1C* enhances the ability of nitrogen uptake and transport, which allocates more nitrogen to grains, thereby increasing the nitrogen utilization efficiency by at least 25%. Over-expression of *OsDREB1C* in rice varieties Nipponbare and Xiushui 134 increases yield by at least 30%. *NGR5* is a positive regulator of rice growth and development in response to nitrogen. The expression level and protein accumulation of NGR5 increases with the increase of fertilizer application [43]. NGR5 interacts with DELLA protein that competitively binds to the GA receptor GID1 protein, which inhibits GA-mediated degradation of the NGR5 protein, thereby increasing its stability. The accumulation of the DELLA protein led to the first “Green Revolution”, achieving high-yield goals, high fertilizer tolerance, and lodging resistance of semi-dwarf plants, but was accompanied by a reduction of nitrogen utilization efficiency. By contrast, the high accumulation of NGR5 and GRF4 proteins did not change the semi-dwarf trait of the “Green Revolution”, but could increase the number of tillers, and promote the absorption and utilization of nitrogen fertilizer, so as to improve the yield and nitrogen utilization efficiency of the existing main cultivars under low nitrogen fertilizer conditions [44]. Therefore, AP2/ERF genes can improve nitrogen utilization efficiency (Figure 4, Table 8), and create excellent germplasms that not only retain the high-yield characteristics of the “Green Revolution” varieties, but also reduce the amount of exogenous nitrogen, which is conducive to cultivate the “less input, more yield”, high nitrogen utilization efficiency and early-maturing rice varieties.

### 3.8. Role of AP2/ERF TFs in Stress Response

Plants can resist biotic and abiotic stresses from different sources by activating various defense mechanisms. AP2/ERF family transcription factors have been found to regulate diverse stress response processes in higher plants, such as abiotic stress (cold, heat, drought, salinity) and biotic stress (insects and pathogens). The mining of AP2/ERF genes and the analysis of regulatory network provides a theoretical basis for breeding rice varieties resistant to biotic and abiotic stresses.

#### 3.8.1. Abiotic Stress Response and Tolerance

Abiotic stresses such as drought, cold damage, high salinity and high temperature seriously affect plant growth and development. AP2/ERF TFs, especially the ERF subfamily members, play a more obvious role in abiotic stress response. *OsDREB1A**, OsDREB1BI**, OsDREB1D* and *OsDREB1E* are positive regulators that resist abnormal temperature. *OsDREB1A* activates the expression of the ion channel gene *CYCLIC NUCLEOTIDE-GATED ION CHANNEL 9 (OsCNGC9)*, which promotes the expressions of genes related to extracellular calcium influx, intracellular calcium concentration and cold stress, and improves the cold tolerance of rice [48]. *OsDREB1BI* can be induced by high and low temperatures and plays a role in cold and heat tolerance of plants [49]. *OsDREB1BI* was transferred into *Arabidopsis*
*thaliana* and the transgenic plants showed tolerance to low and high temperatures. Over-expression of *OsDREB1BI* in tobacco resulted in tolerance to biotic and abiotic stresses [50,51,52]. *OsDREB1D* and *OsDREB1E* were involved in cold and salt stress response. Over-expression of *OsDREB1D* and *OsDREB1E* enhanced cold and high salt resistance in transgenic *Arabidopsis thaliana* [52]. 

Moreover, many AP2/ERF TFs are involved in drought tolerance in rice, such as *OsDREB**s*, *Wax Synthesis Regulatory 2* (*OsWR2*), *OsERF71* and *OsLG3*. *OsDREB1E*, *OsDREB1G* and *OsDREB2B* specifically bound to the C-repeat/DRE elements to regulate drought tolerance. Over-expression of *OsDREB1G* and *OsDREB2B* significantly improves rice drought tolerance, while over-expression of *OsDREB1E* slightly improves rice drought tolerance [53]. *OsDREB1F* is induced by high salt, drought, cold stress and ABA, and specifically binds to DRE/CRT elements. Transgenic rice and *Arabidopsis thaliana* carrying the *OsDREB1F* gene have enhanced resistance to high salt, drought and low temperature [54]. Tian et al. cloned three DREB homologs (*OsDREB1-1*, *OsDREB4-1* and *OsDREB4-2*). In rice seedlings, *OsDREB4-1* was induced by drought and high salt stress, while *OsDREB1-1* and *OsDREB4-2* were constitutively expressed [55]. *OsWR2* positively regulated wax and keratin synthesis and enhanced the tolerance to water scarcity and high temperature [56]. Sensing Ca^2+^ transcription factor 1/2 (SCT1 and SCT2) directly binds to the promoter of *OsWR2* to inhibit its expression, thereby negatively regulating the heat tolerance of rice. SCT1/SCT2 decodes intracellular calcium signals by interacting with calmodulin (CaM). High temperature induces the generation of calcium signal by G protein γ subunit, and high concentration of calcium ions is sensed by CaM and promotes the interaction between CaM and SCT1/SCT2, thereby strengthening the transcriptional inhibitory activity of SCT1/SCT2, which ultimately leads to the rapid down-regulation of *OsWR2* expression at high temperature [57]. Low expression of *OsWR2* reduces wax accumulation, resulting in a heat sensitive phenotype that was unable to resist high temperature [57]. *OsERF71* positively regulates drought tolerance by promoting the formation and expansion of lignification and aerenchyma [58]. Over-expression of *OsERF71* enhances the expression of *OsCCR1*, a key gene for lignin synthesis, thus ensuring the adaptability of root morphology and enhancing drought resistance. *OsLG3* positively regulates rice drought tolerance by inducing the removal of reactive oxygen species [59]. Over-expression of *OsLG3* significantly improves drought tolerance, while inhibition of OsLG3 expression results in increased sensitivity to drought. As a transcription factor that binds to DRE elements, *OsAP211* plays a role in stress signaling [60]. Transgenic seedlings with *OsAP211* inhibition show decreased tolerance to drought stress, while *OsAP211* over-expression plants show slightly increased resistance to drought stress. GUDK mediates drought stress signal transduction through the phosphorylation of OsAP37, leading to the activation of transcription of stress-related genes, which improved stress tolerance, thus increasing yield under drought conditions [61]. Over-expression of *AP37* significantly enhances tolerance to drought, high salt and low temperature stress during vegetative and reproductive growth periods. Under severe drought conditions in the field, transgenic plants significantly improved drought tolerance, and did not affect growth [62,63]. 

*OsAP**23*, *OsEREBP2* and *SERF1* are involved in the salinity tolerance. *OsAP23* is a negative regulator of salt stress response. When exposed to high salt stress, some stress-responsive regulatory genes are significantly induced in the wild type compared with the *OsAP23* over-expression lines [64]. *OsEREBP2* participates in the salt stress response and plays a central role in regulating different abiotic stress responses [64]. Cold stress, ABA, drought and high salt increase the transcription level of the *OsEREBP2* gene [65]. *SERF1* is a salt stress response factor, which amplifies the MAPK cascade signaling pathway activated by reactive oxygen species in the initial response stage of salt stress, and then converts salt-induced signals into an appropriate expression response, thus generating salt tolerance [29]. MAPK5 phosphorylated SERF1 and enhanced the transcriptional activity of *SERF1*. *SERF1* RNAi plants are more sensitive to salt stress, while SERF1 over-expression plants show increased salt tolerance. In addition to regulating spikelet determinacy, *IDS1* also plays an important role in regulating salt tolerance. IDS1 directly binds to the GCC-box in the promoter of the salt stress response genes *LEA1* and *SOS*, and recruits the transcription co-repressor topless related protein TPR1 and histone deacetylase HDA1 to form the IDS1-TPR1-HDA1 module, which inhibits the expression of *LEA1* and *SOS*, thereby negatively regulating the salt tolerance of rice [66]. *OsERF922* negatively regulates salt tolerance in rice [67]. The aboveground parts of *OsERF922* over-expression plants show decreased tolerance to salt stress with a high Na+/K+ ratio. Genomic association and ecotilling method analysis showed that *OsAE115* and *OsAE128* participated in drought stress response [68]. The above information suggests that ERF subfamily genes are involved in high/low temperature, salinity, and drought responses in rice (Figure 5, Table 9).

#### 3.8.2. Biotic Stress Response and Tolerance

ERF subfamily transcription factors play an important role in resistance to blast fungus, *Chilo suppressalis*, rice planthopper and other biological stresses in rice. *OsERF123* is a bacterial blight susceptibility gene. TalB, a transcriptional activator effector from bacterial pathogen *Xanthomonas*, can act on *OsERF123* and *OsTFX1*, leading to higher host sensitivity to bacterial blight fungus *X11-5A* [69]. *OsERF922* is induced by blast fungus and negatively regulates resistance to blast fungus [67]. Knockout of *OsERF922* can enhance resistance to blast fungus, while over-expression of *OsERF922* down regulates the expression of related defense genes and makes plants susceptible to blast fungus. *OsRap2.6* participates in the rice innate immune response through interaction with RECEPTOR FOR ACTIVATED C KINASE 1 (RACK1), which is a positive regulator of innate immunity in rice [70]. *OsRap2.6*-RNAi plants are highly susceptible to blast fungus, whereas *OsRap2.6* over-expression plants show increased resistance to blast fungus [70]. Four benzothiadiazole (BTH)-induced ERF genes, including *OsBIERF1*, *OsBIERF2*, *OsBIERF3* and *OsBIERF4* are found in rice. The expressions of *OsBIERF1*, *OsBIERF3* and *OsBIERF4* are induced by BTH and salicylic acid, which results in a resistance response in rice. Different blast fungi have different abilities to induce the expressions of *OsBIERF1* and *OsBIERF3*, indicating that OsBIERF protein is involved in different signaling pathways of disease resistance responses [71]. *OsERF3* regulates rice resistance to *Chilo suppressalis*. The expression of *OsERF3* is rapidly upregulated by *Chilo suppressalis* feeding, and acts as a central switch in the regulation of plant metabolism to adapt to chewing or sucking insects. *OsERF3* affects the early components of herbivore-induced defense response by inhibiting MAPK inhibitors, modulating plant resistance, and JA, SA, ethylene, H_2_O_2_-mediated signaling pathways [72]. In addition, *OsERF3* is slightly inhibited by brown planthoppers and becomes more sensitive after feeding by brown planthoppers, which might be caused by the inhibition of H_2_O_2_ biosynthesis [72]. These results suggest that ERF subfamily genes play key roles in resistance to rice diseases and insect pests (Figure 5, Table 10).

## 4. Discussion and Prospects

As an important food crop, rice provides the staple food source for half of the world’s population and is one of the key crops studied [73,74]. High yield, high resistance, high quality and green rice have always been the goal of researchers and production development. While the demand for food is increasing, problems such as water shortage or surplus, cold, drought, pests and diseases limit the improvement of rice yield per unit area. As a large plant-specific transcription factor superfamily, AP2/ERF participates in several plant biological and physiological processes and plays a variety of regulatory roles, such as plant reproductive growth, vegetative growth and various stress-induced stress responses, and plays important roles in determining crop yield and adapting to stress. In this review, we summarize the classification, features, function, regulatory mechanisms, and potential applications of AP2/ERF transcription factors in rice. Although some AP2/ERF regulators and their molecular roles have been identified, their regulatory networks remain fragmented. In fact, the research on AP2/ERF transcription factors in rice is still at the stage of single gene function analysis, and the research on its underlying mechanisms is carried out in the laboratory rather than in the field. One future challenge is to explore the upstream and downstream components of AP2/ERF transcription factors and the possible connections between regulatory pathways under field conditions. The other challenge is to solve the trade-off effect among high yield, high resistance and green, organic production. With our greatly increased knowledge of the molecular mechanisms that determine these traits, the combination of beneficial alleles controlling these characters will help breeders to develop desired rice varieties with high yield, high resistance, and green, organic production.

## Figures and Tables

**Figure 1 ijms-23-12013-f001:**
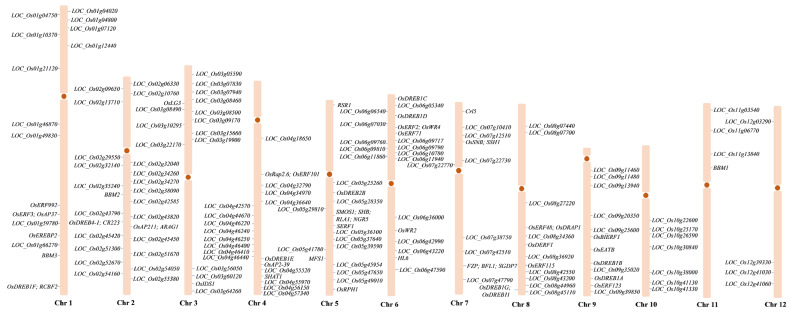
The distribution of AP2/ERF genes on chromosomes in rice.

**Figure 2 ijms-23-12013-f002:**
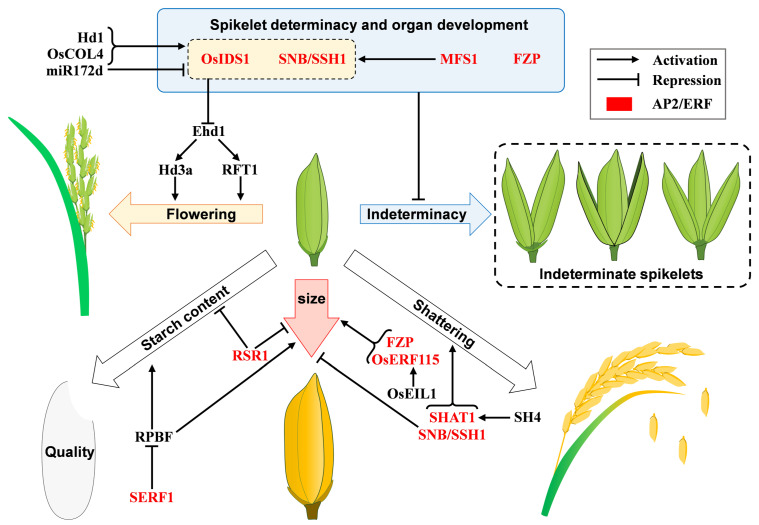
Schematic representation of spikelet development, flowering time and grain properties mediated by AP2/ERF transcription factors in rice. *O**sIDS1*, *SNB/SSH1*, *MFS1* and *FZP*, which are mentioned in Section 3.1, inhibit spikelet indeterminacy. Of these, *OsIDS1* and *SNB/SSH1* are mentioned in Section 3.2 to inhibit flowering. In Section 3.3, *SHAT1* and *SNB/SSH1* are mentioned to promote grain shattering. *FZP* and *OsERF115* promote grain size, while *SNB/SSH1*, *RSR1* and *SERF1* inhibit grain size. *SERF1* inhibits starch content, and *RSR1* affect amylose content and amylopectin structure. The AP2/ERF transcription factors and other regulators are shown in red and black, respectively.

**Figure 3 ijms-23-12013-f003:**
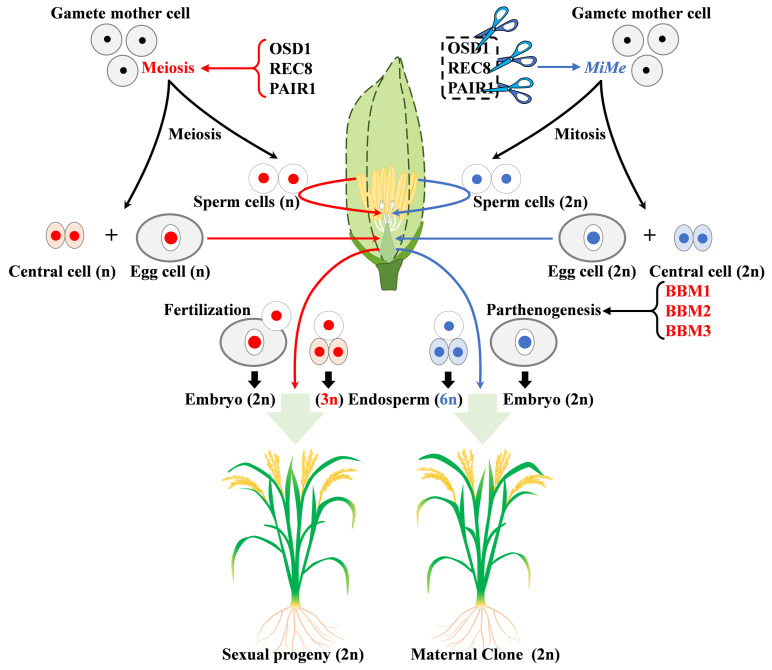
Schematic representation of sexual reproduction and synthetic apomixis regulated by *MiMe* and BBMs in rice. The genes in this figure are described in Section 3.4. Left is the sexual reproduction process in which the germ cells form haploid gametes through processes such as meiosis and double fertilization that produce a diploid (2n) embryo and a triploid (3n) endosperm. Apomixis on the right consists of two main modifications: one is the knockdown of *OSD1*, *REC8* and *PAIR1*, which converts meiosis into mitosis (a technique called *MiMe*), resulting in 2n egg, sperm cells and central cells. The second is parthenogenesis induced by BBMs (AP2/ERF transcription factors), which converts (2n) egg cells into embryos and produces (6n) endosperm.

**Figure 4 ijms-23-12013-f004:**
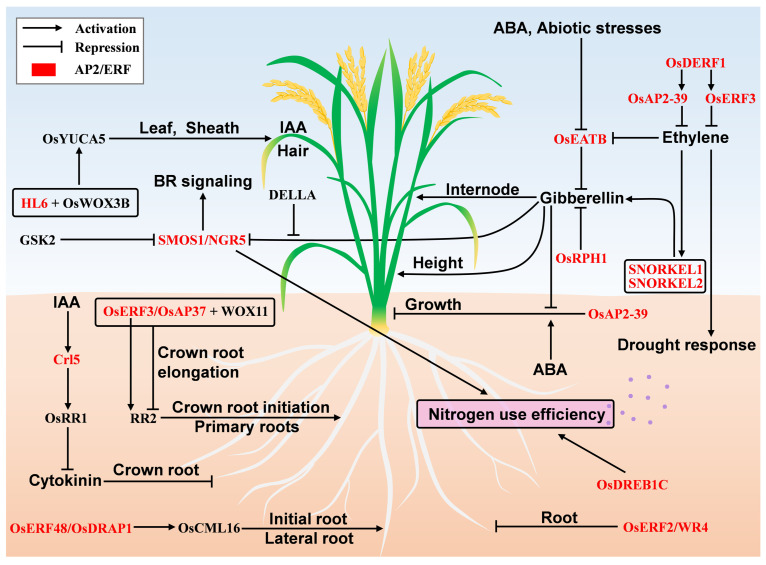
Schematic representation of development, phytohormones and nutrient use efficiency regulation mediated by AP2/ERF transcription factors in rice. In Section 3.5, *O**sERF3/O**sAP37*, *C**rl5* and *O**sERF48/O**sDRAP1* are described as affecting root initiation and formation. In Section 3.6, *HL6, OsEATB, OsERF2/WR4, OsRPH1, SMOS1/NGR5, SNORKEL1, SNORKEL2, OsDERF1, OsAP2-39* and *OsERF3* are described to regulate various growth processes of plants by participating in phytohormone pathways. *SMOS1/NGR5* and *OsDREB1C* promote rice nitrogen use efficiency, as described in Section 3.7. The AP2/ERF transcription factors and other regulators are shown in red and black, respectively. Abbreviations: indole-3-acetic acid, IAA; abscisic acid, ABA; gibberellin, GA; brassinosteroid, BR.

**Figure 5 ijms-23-12013-f005:**
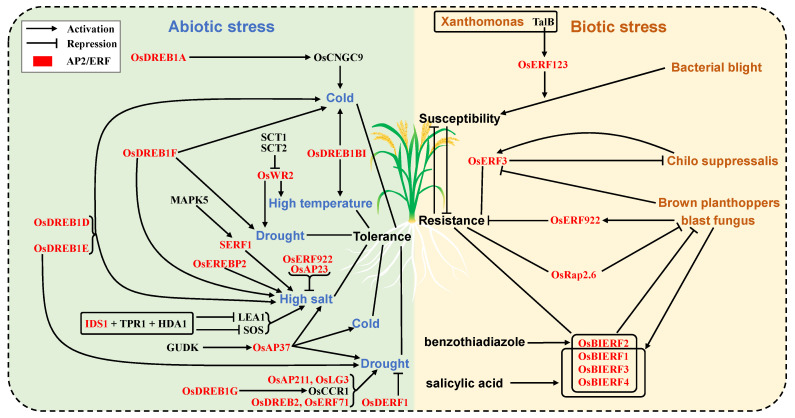
Schematic representation of AP2/ERF transcription factors involved in abiotic and biotic stress regulation in rice. Left are the AP2/ERF regulatory networks of cold, high temperature, high salt and drought tolerance in rice; these factors are discussed in Section 3.8.1. *O**sDREB1A/D/E/F, O**sDREB1BI* and *O**sAP37* promote rice cold tolerance. *O**sDREB1BI* and *O**sWR2* promote rice high temperature tolerance. *O**sDREB1E/F/G, O**sDREB2, O**sWR2, O**sAP37, O**sAP211, O**sLG3* and *O**sERF71* promote rice drought tolerance, while OsDERF1 inhibits drought tolerance. *O**sDREB1D/E/F, SERF1, O**sEREBP2, O**sAP37* and *IDS1* promote rice high salt tolerance, while *O**sERF922* and *O**sAP23* inhibit high salt tolerance. On the right, AP2/ERF genes participate in the regulation of disease and insect resistance in rice; these factors are discussed in Section 3.8.2. TalB, which is a transcriptional activator effector in Xanthomonas, promotes rice susceptibility to bacterial blight fungus *X11-5A* by promoting *OsERF123*. *OsERF3* is activated by *Chilo suppressalis* attack, which improves rice resistance against chewing herbivores but is slightly suppressed by brown planthoppers attack. *OsRap2.6* promotes rice blast fungus resistance, while *OsERF922* inhibits rice blast fungus resistance. *OsBIERF1/2/3/4* is involved in the rice blast defense response, and among them *OsBIERF1/2/3/4* can be induced by benzothiadiazole, and *OsBIERF1/3/4* can be induced by salicylic acid and blast fungus. The AP2/ERF transcription factors and other regulators are shown in red and black, respectively.

**Table 1 ijms-23-12013-t001:** Classification and quantity of AP2/ERF transcription factors in rice.

Classification	AP2	ERF	RAV	Soloist
Number	24	136	5	0

**Table 2 ijms-23-12013-t002:** The genes associated with spikelet determinacy and organ development in rice.

Gene Name	Gene Family	Gene Locus	Gene Function	Reference(s)
*FZP; BFL1; SGDP7*	ERF	*LOC_Os07g47330*	spikelet development	[19,20]
*OsSNB; SSH1*	AP2	*LOC_Os07g13170*	spikelet development	[21]
*OsIDS1*	AP2	*LOC_Os03g60430*	spikelet development	[22]
*MFS1*	ERF	*LOC_Os05g41760*	spikelet development	[18]

**Table 3 ijms-23-12013-t003:** Genes associated with flowering in rice.

Gene Name	Gene Family	Gene Locus	Gene Function	Reference
*OsIDS1*	AP2	*LOC_Os03g60430*	flowering	[22]
*OsSNB; SSH1*	AP2	*LOC_Os07g13170*	flowering	[23]

**Table 4 ijms-23-12013-t004:** Genes associated with grain properties in rice.

Gene Name	Gene Family	Gene Locus	Gene Function	Reference(s)
*OsSNB; SSH1*	AP2	*LOC_Os07g13170*	grain size; grain shattering	[24,25]
*SHAT1*	AP2	*LOC_Os04g55560*	grain shattering	[26]
*OsERF115*	ERF	*LOC_Os08g41030*	grain size	[27]
*FZP; BFL1; SGDP7*	ERF	*LOC_Os07g47330*	grain size, grain number	[19,28]
*SERF1*	ERF	*LOC_Os05g34730*	grain size; grain quality	[29]
*RSR1*	AP2	*LOC_Os05g03040*	grain quality	[30]

**Table 5 ijms-23-12013-t005:** Genes associated with embryogenesis in rice.

Gene Name	Gene Family	Gene Locus	Gene Function	Reference
*BBM1*	AP2	*LOC_Os11g19060*	embryonic development	[31]
*BBM2*	AP2	*LOC_Os02g40070*	embryonic development	[31]
*BBM3*	AP2	*LOC_Os01g67410*	embryonic development	[31]

**Table 6 ijms-23-12013-t006:** Genes associated with root development in rice.

Gene Name	Gene Family	Gene Locus	Gene Function	Reference(s)
*Crl5*	AP2	*LOC_Os07g03250*	root initiation and formation	[33,34]
*OsERF48; OsDRAP1*	ERF	*LOC_Os08g31580*	root initiation and formation	[35]
*OsERF3; OsAP37*	ERF	*LOC_Os01g58420*	root initiation and formation	[36]

**Table 7 ijms-23-12013-t007:** Genes associated with hormone regulation in rice.

Gene Name	Gene Family	Gene Locus	Gene Function	Reference(s)
*OsDERF1*	ERF	*LOC_Os08g35240*	drought tolerance	[37]
*OsEATB*	ERF	*LOC_Os09g28440*	internode development	[40]
*HL6*	ERF	*LOC_Os06g44750*	epidermis hair development	[46]
*OsERF2; OsWR4*	ERF	*LOC_Os06g08340*	root development and formation	[39]
*OsRPH1*	ERF	*LOC_Os05g49700*	internode development	[41]
*SMOS1; SHB; RLA1; NGR5*	ERF	*LOC_Os05g32270*	signal transduction	[43,44,45]
*OsAP2-39*	ERF	*LOC_Os04g52090*	signal transduction	[42]
*SNORKEL1*	ERF	*AB510478*	internode development	[38]
*SNORKEL2*	ERF	*AB510479*	internode development	[38]

**Table 8 ijms-23-12013-t008:** Genes associated with nutrient use in rice.

Gene Name	Gene Family	Gene Locus	Gene Function	Reference
*SMOS1; SHB; RLA1; NGR5*	ERF	*LOC_Os05g32270*	nutrient use	[43]
*OsDREB1C*	ERF	*LOC_Os06g03670*	photosynthetic efficiency; nitrogen use	[47]

**Table 9 ijms-23-12013-t009:** Genes associated with abiotic stress in rice.

Gene Name	Gene Family	Gene Locus	Gene Function	Reference(s)
*OsDREB1A*	ERF	*LOC_Os09g35030*	cold tolerance	[48]
*OsDREB1B*	ERF	*LOC_Os09g35010*	cold tolerance; heat tolerance;	[49,50,51,52]
*OsDREB1D*	ERF	*LOC_Os06g06970*	cold tolerance; salinity tolerance	[52]
*OsDREB1E*	ERF	*LOC_Os04g48350*	drought tolerance	[52]
*OsDREB1G; OsDREB1I*	ERF	*LOC_Os08g43210*	drought tolerance	[53]
*OsDREB2B*	ERF	*LOC_Os05g27930*	drought tolerance	[53]
*OsDREB1E*	ERF	*LOC_Os04g48350*	drought tolerance	[53]
*OsDREB1F; RCBF2*	ERF	*LOC_Os01g73770*	salinity tolerance; drought tolerance; temperature tolerance	[54]
*OsDREB1-1; CR350*	ERF	*LOC_Os04g48350*	salinity tolerance; drought tolerance	[55]
*OsDREB4-1; CR223*	ERF	*LOC_Os02g43940*	salinity tolerance; drought tolerance	[55]
*OsWR2*	ERF	*LOC_Os06g40150*	temperature tolerance	[56,57]
*OsERF71*	ERF	*LOC_Os06g09390*	drought tolerance	[58]
*OsLG3; OsERF62; OsRAF*	ERF	*LOC_Os03g08470*	drought tolerance	[59]
*OsAP211; ARAG1*	ERF	*LOC_Os02g43970*	drought tolerance	[60]
*OsAP37*	ERF	*LOC_Os01g58420*	drought tolerance	[61,62,63]
*OsAP23*	ERF	*LOC_Os03g05590*	salinity tolerance	[64]
*OsEREBP2*	ERF	*LOC_Os01g64790*	salinity tolerance; drought tolerance; temperature tolerance	[65]
*SERF1*	ERF	*LOC_Os05g34730*	salinity tolerance	[29]
*OsIDS1*	AP2	*LOC_Os03g60430*	salinity tolerance	[66]
*OsERF922*	ERF	*LOC_Os01g54890*	salinity tolerance; drought tolerance	[67,68]

**Table 10 ijms-23-12013-t010:** Genes associated with biotic stress in rice.

Gene Name	Gene Family	Gene Locus	Gene Function	Reference
*OsERF123*	ERF	*LOC_Os09g39810*	bacterial blight	[69]
*OsERF922*	ERF	*LOC_Os01g54890*	blast fungus	[67]
*OsRap2.6; OsERF101*	ERF	*LOC_Os04g32620*	blast fungus	[70]
*OsBIERF1*	ERF	*LOC_Os09g26420*	blast fungus	[71]
*OsERF3; OsAP37*	ERF	*LOC_Os01g58420*	chilo suppressalis; brown planthoppers	[72]

## Data Availability

Not applicable.

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
