# Peer review of "Molecular Events of Rice AP2/ERF Transcription Factors"

_ijms, 2022, doi:10.3390/ijms231912013_

Round 1

Reviewer 1 Report

1. The title of manuscript

The manuscript is called "Molecular events and breeding potential of rice AP2/ERF transcription factors", but there is very little about breeding potential in it. If the authors consider it important to emphasize this, then they should either make a separate section on the topic, or describe this problem well in the Discussion (now it contains only 1-2 phrases about this).

2. Introduction

Introduction is very small informative, it should be expanded.

3. Section 3:

The description of gene functions in various developmental processes in the section 3 is poorly systematized, the data is simply "piled up", the abbreviations of the genes are not deciphered, the molecular functions of their products are not enough described (this is especially noticeable in the section 3.5). The text is not divided into paragraphs, it is hard to read. Not all of genes which were listed in the subsections of section 3 belong to the ERF family or somehow interact with the ERF, then the reason why they are mentioned in the text is not very clear.

4. Subsection 3.1. Spikelet determinacy and organ development

Here you need to write a little about the spikelet development especially since the section deals with spikelet meristem, etc.

5. Subsection 3.6. Hormone regulation

The strange name of the section - no specifics. Functions of AP2/ERF TFs in interaction with different hormones are mixed into one heap.

6. Figures 2-5.

These figures must illustrate the certain subsecrtions, but not all genes presented in the figures are in the text of corresponding subsections and vice versa.

7. Table 9

The second column "moved out", as a result, it is not clear which gene belongs to which family.

8. Everywhere in the manuscript:

·         Abbreviations of all genes must be spell out at the first mention.

·         The osids1/snb double mutant (line 87), bbm1/bbm2/bbm3 triple mutant (line 133), etc.: Double mutants shouldn't be written like that.

·         Arabidopsis Thaliana - Species name should be should be written in lower case

·         Numerous typos (lemma/plaea-like – line 89, highresistance – line 451, etc.)

Line 25-26: In plants, about 60 kinds of transcription factor families were identified,

Very strange sentence: “Kinds of families”

Line 28-29: AP2/ERF is one of the oldest and largest transcription factor families in plants

Evolutionary oldest?

Line 29-31:

It was first isolated from Arabidopsis Thaliana by Jofuku in 1994 and was widely involved in a series of plant growth and development processes, including plant growth

Need to change the phrase:  two times “plant growth”

Line 35-36:

In the plant kingdom, the AP2/ERF family is a large family of transcription factors
with a conserved AP2/ERF binding domain

Maybe “DNA binding domain”?

Line 39-41:

ERF family proteins have only one AP2/ERF domain, which can be subdivided into two subfamilies, ERF and CBF/DREB according to the binding DNA sequence.

Maybe “sequence of DNA binding domain”?

Line 42-44

The proteins encoded by ERF subfamily genes bind to AGCCGCC core motifs, while CBF/DREB subfamily members contain the cis-acting elements A/GCCGAC recognized by C-repeat

Understood nothing. What is C-repeat?  Why does this C-repeat recognize cis-acting elements A/GCCGAC contained in the transcription factor?

Line 46-48

These proteins are negative regulators of growth and development in Arabidopsis, and play key regulatory roles in potato defense pathways

What is “potato defense system”?

Line 48-51

Phylogenetic analysis of AP2/ERF TFs in rice showed that alanine 14 (Ala-14) and aspartic acid 19 (Asp-19) of ERF subfamily genes (PROTEINS???) were conserved among the members containing a single AP2/ERF domain, while CBF/DREB subfamily gene (PROTEIN???) was (CONTAINS?) valine (Val-14) at 14th and glutamine (Glu-19) at 19th

Line 54-55

It was also found that there is histidine in each domain of the two-domain proteins, but not in the single-domain proteins

Histidine in some conserved positions?

Line 71-72

LHS1/OsMADS1 in the SEP subfamily and AP2/ERF family genes (Table 2).

The SEP subfamily is neither in the text above nor in Table 2.

Line 408-414

Who is Xanthus (it means Xanthomonas?) and what does it mean RACK1?

Author Response

  1. The title of manuscript

The manuscript is called "Molecular events and breeding potential of rice AP2/ERF transcription factors", but there is very little about breeding potential in it. If the authors consider it important to emphasize this, then they should either make a separate section on the topic, or describe this problem well in the Discussion (now it contains only 1-2 phrases about this).

ResponseThanks for your suggestion, we have revised the title.

  1. Introduction

Introduction is very small informative, it should be expanded.

Response: Thanks for your suggestion, we have expanded the introduction.

  1. Section 3:

The description of gene functions in various developmental processes in the section 3 is poorly systematized, the data is simply "piled up", the abbreviations of the genes are not deciphered, the molecular functions of their products are not enough described (this is especially noticeable in the section 3.5). The text is not divided into paragraphs, it is hard to read. Not all of genes which were listed in the subsections of section 3 belong to the ERF family or somehow interact with the ERF, then the reason why they are mentioned in the text is not very clear.

Response: Thanks for your suggestion. We have made the more specific description of gene functions. We also provided the full name of the genes. And we have adjusted the contents of some complex paragraphs to make them read easily. Non-AP2/ERF genes interact with the AP2/ERF genes to perform corresponding functions together. And, AP2/ERF genes are also described briefly.

Subsection 3.1. Spikelet determinacy and organ development

Here you need to write a little about the spikelet development especially since the section deals with spikelet meristem, etc.

Response: Thanks for your suggestion. We have described more details for spikelet development.

  1. Subsection 3.6. Hormone regulation

The strange name of the section - no specifics. Functions of AP2/ERF TFs in interaction with different hormones are mixed into one heap.

Response: Thanks for your suggestion. We have revised the section and the title. And, AP2/ERF genes induce hormone response by activating target genes or regulating various growth processes of rice as response factors.

  1. Figures 2-5.

These figures must illustrate the certain subsections, but not all genes presented in the figures are in the text of corresponding subsections and vice versa.

Response: Thank you. We have revised it in the text and figure legends.

  1. Table 9

The second column "moved out", as a result, it is not clear which gene belongs to which family.

ResponseThanks for your suggestion, DREB subfamily also belongs to ERF subfamily, we have revised the content of second column in table 9.

  1. Everywhere in the manuscript:

1、Abbreviations of all genes must be spell out at the first mention.

2、The osids1/snb double mutant (line 87), bbm1/bbm2/bbm3 triple mutant (line 133), etc.: Double mutants shouldn't be written like that.

  • 3、Arabidopsis Thaliana - Species name should be should be written in lower case

4、Numerous typos (lemma/plaea-like – line 89, highresistance – line 451, etc.)

5、Line 25-26: In plants, about 60 kinds of transcription factor families were identified,

Very strange sentence: “Kinds of families”

6、Line 28-29: AP2/ERF is one of the oldest and largest transcription factor families in plants

Evolutionary oldest?

7、Line 29-31:

It was first isolated from Arabidopsis Thaliana by Jofuku in 1994 and was widely involved in a series of plant growth and development processes, including plant growth

Need to change the phrase:  two times “plant growth”

8、Line 35-36:

In the plant kingdom, the AP2/ERF family is a large family of transcription factors with a conserved AP2/ERF binding domain

Maybe “DNA binding domain”?

9、Line 39-41:

ERF family proteins have only one AP2/ERF domain, which can be subdivided into two subfamilies, ERF and CBF/DREB according to the binding DNA sequence.

Maybe “sequence of DNA binding domain”?

10、Line 42-44

The proteins encoded by ERF subfamily genes bind to AGCCGCC core motifs, while CBF/DREB subfamily members contain the cis-acting elements A/GCCGAC recognized by C-repeat

Understood nothing. What is C-repeat?  Why does this C-repeat recognize cis-acting elements A/GCCGAC contained in the transcription factor?

11、Line 46-48

These proteins are negative regulators of growth and development in Arabidopsis, and play key regulatory roles in potato defense pathways

What is “potato defense system”?

12、Line 48-51

Phylogenetic analysis of AP2/ERF TFs in rice showed that alanine 14 (Ala-14) and aspartic acid 19 (Asp-19) of ERF subfamily genes (PROTEINS???) were conserved among the members containing a single AP2/ERF domain, while CBF/DREB subfamily gene (PROTEIN???) was (CONTAINS?) valine (Val-14) at 14th and glutamine (Glu-19) at 19th

13、Line 54-55

It was also found that there is histidine in each domain of the two-domain proteins, but not in the single-domain proteins

Histidine in some conserved positions?

14、Line 71-72

LHS1/OsMADS1 in the SEP subfamily and AP2/ERF family genes (Table 2).

The SEP subfamily is neither in the text above nor in Table 2.

15、Line 408-414

Who is Xanthus (it means Xanthomonas?) and what does it mean RACK1?

Response: Thanks for your suggestion. We have corrected the errors mentioned above.

Reviewer 2 Report

1. The conclusion of the abstract requires authors to give a relatively clear summary, what regulatory mechanism does AP2/ERF in rice clarify?

2. The first and second pages of this manuscript are well written, and the last paragraph is also very concise, but from page 3 to the penultimate paragraph, I feel a lot of descriptive information is not very clear, although authors try to use diagrams or tables to make this part of the content clearer. I would like to recommend that authors summarize a conclusion in the last sentence of each paragraph, so that readers will be better grasp the key content of this manuscript.

3. A large number of gene abbreviations do not have any full names. It is recommended that authors list all the abbreviations in a table and write the full names, which will be more conducive to the understanding and signing of the readers to this manuscript.

Author Response

  1. The conclusion of the abstract requires authors to give a relatively clear summary, what regulatory mechanism does AP2/ERF in rice clarify?

ResponseThanks for your criticism. We have summarized the regulatory mechanism of AP2/ERF transcription factors in the abstract.

  1. The first and second pages of this manuscript are well written, and the last paragraph is also very concise, but from page 3 to the penultimate paragraph, I feel a lot of descriptive information is not very clear, although authors try to use diagrams or tables to make this part of the content clearer. I would like to recommend that authors summarize a conclusion in the last sentence of each paragraph, so that readers will be better grasp the key content of this manuscript.

Response: Thanks for your suggestion. We have added the conclusion in each paragraph to make it more systematic.

  1. A large number of gene abbreviations do not have any full names. It is recommended that authors list all the abbreviations in a table and write the full names, which will be more conducive to the understanding and signing of the readers to this manuscript.

Response: Thanks for your suggestion. We have provided the full name for the abbreviations of the genes.

Round 2

Reviewer 2 Report

I recommended to accept this manuscript.